# Comparative Analysis on Urban Flood Countermeasures Based on Life Cycle Thinking: A Comparison between Enhancing of Drainage Capacity Project and Sponge City

**Xuezhou Fan \*** and **Toru Matsumoto**

Faculty of Environmental Engineering, University of Kitakyushu, Fukuoka 808-0135, Japan; matsumoto-t@kitakyu-u.ac.jp
\* Correspondence: fanxuezhou@163.com; Tel.: +81-70-2347-6018

**Abstract:** The recent increase in rainstorm waterlogging disasters has acutely threatened sustainable urban development in China. Traditional strategy to solve this problem is drainage capacity enhancing projects, which aims at enlarging the discharge of water. Recently, there is a new countermeasure emerged in Chinese cities: 'Sponge City', which aims at enlarging the absorption of water by increasing the curves of urban land. This article endeavours to make a comparison between these two countermeasures by building a framework to design and analyze the private or social costs of two projects which have the same rainwater control capacity. Finally, we have come to the conclusion that only considering initial cost, Sponge City unit cost is much more than drainage capacity enhancing project unit cost; considering external cost, Sponge City is not only competitive from an economical perspective, from an environmental perspective Sponge City is also competitive.

**Keywords:** urban flood countermeasure; life cycle; Sponge City

## 1. Introduction

From the 1970s until now, China has experienced a rapid urbanization increasing phase. In such a rapid process of urbanization, land use change from natural landscape (e.g., green spaces, vegetation areas, forestry, and soil surfaces) to urban land-use (e.g., commercial, residential, and industrial) has been unprecedented. In this way then, a reduction in permeable surfaces and rainfall infiltration has resulted in a lower retention capacity for storm water in urban areas and less recharge of underlying groundwater. Climate change and increasing of intense storms may further enhance these fluctuations and flood risks. Meanwhile, the upgrading of urban drainage systems lags behind these increasing risks. Many urban drainage systems in Chinese cities operate in exceedance of conditions for which they were initially designed [1]. For instance in Shenzhen, the urban runoff discharge has been increased by 12.9% from the 1980s to the 2000s [2]; in Harbin, municipal drainage pump stations provide a drainage capacity of 135 m³/s, however according to standards to protect against moderate rainfall (25 mm/h), the city requires a drainage capacity of 185 m³/s [3]. As a result, a number of cities have suffered recurrent flooding as their outdated drainage systems fail to cope with the rainstorms. For instance, on 21 and 22 July 2012, Beijing suffered the most severe storm and flood disaster in 61 years. The average rainfall in the city was 170 mm, and that in the city center was 215 mm. The storm caused 79 deaths, 10,754 cars were submerged, and 10,660 houses collapsed. In 2014, Guangzhou City in Guangdong Province suffered an extreme rainfall event, which caused many streets to become inundated and led to severe transport interruptions [4]. In 2016, Wuhan City in Hubei Province encountered a heavy rainstorm, in which 14 people died, 1 person went missing, and

32,160 ha of crops and vegetables were destroyed [5,6]. Data shows that in over 90% of Chinese cities, the design of urban flooding mitigation measures are linked to traditional engineering infrastructure (i.e., floodgates, concrete infrastructure, and oversized drains) that aim to drain urban discharge as quickly as possible to downstream outlets (larger rivers, lakes, or coasts), however this approach is inflexible in accommodating unplanned urban expansion and increases in impervious urban land surfaces and can have a heavy environmental impact in material production and project construction phases. The 'Sponge City' concept aims to transform the urban planning process whilst promoting the conservation and creation of greener landscapes in urban areas and will engage more effectively with future land-use and spatial planning and also improves urban ecosystem diversity and social wellbeing. A 'Sponge City' refers to Integrated Urban Water Management (IUWM), which combines the management of water supply, groundwater, wastewater, and storm water in cities, which is not a new concept. The term 'Sustainable Urban Drainage Systems' (SUDS), used to describe stormwater technology, was coined first by Jim Conlin of Scottish Water (October 1997) and a sustainable drainage triangle (quantity, quality, habitat) was initially set out by D'Arcy (1998) [7]. In practical application, Water Sensitive Urban Design (WSUD) in Australia, and Low Impact Development (LID) in the US are similar concepts to Sponge City [8]. Some prevalent technologies are applied in Sponge City construction—such as green roofs, rain gardens, water-permeable pavement, and vegetative swales.

There are many studies on urban drainage infrastructure social cost assessment. By identifying the eight most important social cost categories, Matthews et al. calculated two pipe replacement projects in California US and Kessel-Dorp, Belgium separately, and concluded that for two cases the inclusion of social costs in the project cost estimate could make trenchless technology more advantageous in comparison with open-cut construction, and they also mentioned that these results are especially true for high density urban areas [9]. Iwashita et al. calculated appropriate renewal time by considering breakage risk and social costs, which includes internal costs and external costs, and concluded that the renewal time has been extended when considering social costs, compared to only considering internal costs [10]. Wang et al. calculated and compared directed cost and social cost of drainage pipe rehabilitation projects in open an cut way and non-cut way respectively, and arrived at the conclusion that considering indirect costs, social costs of rehabilitation projects of the non-cut way are competitive with the cut way in the Shanghai urban area [11]. There are also many studies on social cost and social benefit analysis on Sponge City technologies. Bianchini et al. calculated personal and social costs and benefits of green roof construction in life cycle, and showed the effectiveness of building green roofs in urban storm water management [12]. Lin et al. showed that, in Shanghai city, after building sponge city facilities in target residential communities, the accumulated carbon sink after 18.8 years can equal the total amount of carbon emissions. Additionally, in the subsequent lifespan, these sponge city facilities can function as carbon emission reduction system by absorbing greenhouse gases. In the last 11 years of a 30-year life cycle, the amount of net carbon emission absorbing benefit was calculated to be 25,407 kg $CO_2$ eq/y [13]. Fan et al. calculated that by building area of 592 thousand $m^2$ extensive green roofs (as per the Sponge City project) in a target area, urban waterlogging can be mitigated, and the benefit of these roofs is exceeds their costs from a life cycle perspective, which demonstrates green roofs' effectiveness [14].

Because the traditional engineering infrastructure approach is inflexible in accommodating unplanned urban expansion, evaluation of replacing some sewer drainage systems with green infrastructure (Sponge City) becomes comparatively practical. However, many studies on social cost analysis of urban drainage infrastructure and Sponge City technologies have been done individually, the comparative analysis between them based on same capacity of storm water reduction is insufficient. This paper tries to build a framework to design and analyze these two projects (two approaches) to make a comparative analysis of them from a case study in life cycle and propose the measuring method when making a project selection decision.

## 2. Method

### 2.1. Design and Cost Estimating Basis of This Study

Drainage capacity enhancing infrastructure design involves a large-size combined sewerage pipes replacement project, which is constructed in open-cut way; Sponge City construction includes green roofs, rain gardens, water-permeable pavement, and vegetative swales; involving several technologies and material, this building progress changes with different technologies. Combined sewerage pipes are designed based on "Code for design of outdoor wastewater engineering" (GB50014-2006) (hereinafter called "Code") [15], and its initial cost estimate is based on "Quota for National unified municipal engineering budget, book 6" (1999) (hereinafter called "Quota") [16]. Sponge City is designed based on "Sponge City construction technical manual" (hereinafter called 'Manual') [17], and its initial cost is estimated based on "Indices for investment estimate of Sponge City construction" (hereinafter called 'Indices') [18]. From "Quota" which was published in 1999 (with no new version), the cost estimates in this study will consider inflation rates and existing studies in concert.

### 2.2. Boundary and Inventory for the Two Projects

Determining a reasonable social cost (including environmental cost) accounting boundary is an essential step in studying the cost of drain replacement project and Sponge City project, and the results vary largely according to the determination of inventory. This study uses a LCA (Life Cycle Assessment) approach to analyze the social cost in the whole process of drain replacement project and Sponge City project, including phases of material production, construction, operation, and disassembly. The life span of Sponge City and combined sewerage pipes are set as 50 years. Boundary and inventory for both projects are shown in Figures 1 and 2. Additionally, all the costs are introduced in RMB (RMB: Chinese currency, US$1 = RMB 6.83 present exchange rate).

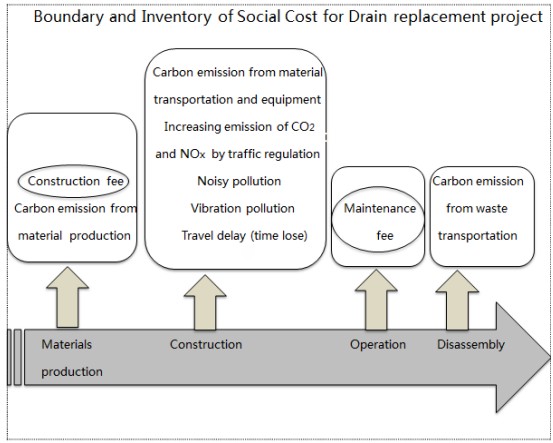

**Figure 1.** Boundary and Inventory of social cost for drain replacement project.

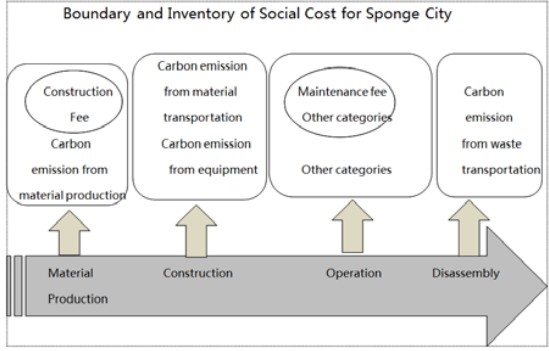

**Figure 2.** Boundary and Inventory of social cost for Sponge City.

## 3. Case Study

### 3.1. Study Area

The case mentioned in Zhang et al. and Fan et al. is studied in this paper. The district of Nangang in Harbin city lies in mid-south section of Harbin, Heilongjiang province, in Northeastern China. Annual average precipitation is 569.1 mm. Most of the drainage system in Harbin is composed of confluence pipe and was first established in 1904, and the main part of the city has 1161 km of municipal, sewer drains, 24,483 check wells, 14,381 rain wells, and a drainage network penetration rate of 66.17%; drainage system design conveyance capacity was 18 mm/h. Harbin's municipal drainage pump stations provide a drainage capacity of 135 m$^3$/s. According to standards to protect against moderate rainfall (25 mm/h), the city requires a drainage capacity of 185 m$^3$/s. Therefore, serious waterlogging occurs in the city with only moderate precipitation. The area between Kuancheng Street and Dacheng Street in Nangang District (Figure 3), consisting of 63 buildings and 7 main impervious roads, is one of the severe flooding areas. According to "Harbin water supply and drainage industry development strategic planning", Sponge City or an upgraded drainage system project will be implemented from 2020 until 2025 to mitigate increasing urban flooding in Harbin City.

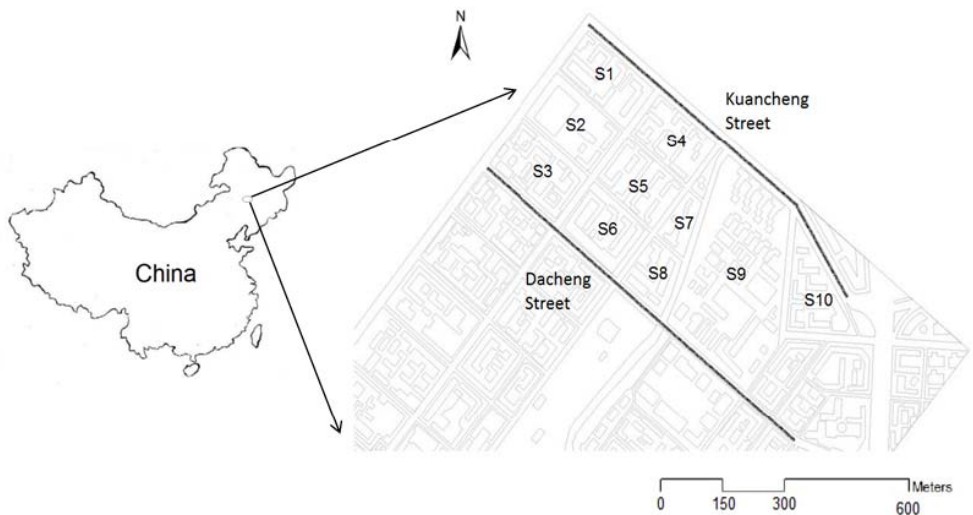

**Figure 3.** Location of target area and planning for drain replacement project (target area was divided into 10 subcatchments, S1, S2, . . . ,S10).

Firstly, we assume that a combined drain replacement project is to be implemented underneath Kuancheng Street and Dacheng Street (all residential pipes near to these two streets are retained). Based on 'Code' (p. 21), rainwater runoff design discharge $Q_s$(L/s) can be calculated by

$$Q_s = q.\psi.F \qquad (1)$$

$q$: Designed rainstorm intensity (L/ (s. hm$^2$))
$\psi$: Runoff coefficient, which is introduced as 0.45
$F$: Catchment area (ha), is estimated as 11.95 ha.
$q$: (L/(s.hm$^2$)) is calculated by rainstorm intensity formula 'Code' (p. 22)

$$q = \frac{167A_1(1 + ClgP)}{(t + b)^n} \qquad (2)$$

$t$: Rainfall duration (min)
$P$: Designed return period (year)

$A_1$, $C$, $b$, and $n$ are regional parameters, which embody the empirical observation of an area or city. Data draw on the monthly maximum 60-min rainfalls in 2017 and 2018 in Harbin City and the maximum one is obtained as 50.40 mm (Data collected from Harbin Meteorological Bureau). Owning to the limitations of data, rainstorm intensity was calculated as hourly average rainstorm intensity and rainfall duration here was set as 60 minutes and return period was set as 2 years. Four parameters are introduced as 17.932, 17.036, 11.770, and 0.880 respectively (refer to "Water supply and drainage design manual", book 5). Domestic sewage runoff design discharge $Q_h$(L/s) is the sum of sewage runoff design discharge from subcatchment 1 to subcatchment 10 (Figure 3), in which the sewage of subcatchments 1, 2, 4, 5, 7, and 10 flow to the Kuancheng combined drain and the sewage of subcatchments 3, 6, 8, and 9 flows to the Dacheng combined drain. Population density of target area is 315/ha and sewage quota is 100 L/day per capita, designed pipe's interception ratio is 3 and its buried depth and slope are the same with exiting drain. Additionally, because of lack of water quality data, in this combined sewerage pipe designing work, wastewater quality designing is not considered. Finally, both drain diameters are calculated as 1200 mm. The diameter of existing pipes is 600 mm, which may be the reason that this area became severely waterlogging district. The drainage capacity before and after replacement project are calculated as 0.283 m$^3$/s and 1.13 m$^3$/s flow speed in drain were designed as 1 m/s and both are in full flow condition. Therefore, the drainage capacity of enhancing discharges is calculated as 0.847 m$^3$/s.

There are 10 subcatchments in target area and technologies as green roofs and water storage sand ponds are to be applied to reduce overflow in storm. The inventory of main materials used in facilities construction in this Sponge City project is sand, concrete, and steel, which was obtained from engineering designers and actual construction organizations (for cost reducing, old original plastic pipes were reused). In order to ensure that the designed Sponge City has the same runoff 'absorbing capacity', 6.05 ha container green roofs ('absorbing capacity' is 3049 m$^3$ when maximum hourly precipitation reached) are designed on top of buildings nearby and six sand ponds (size is 500 m$^3$), are designed to build into subcatchments 2, 6, 8, 9, and 10 respectively (there are two ponds are built in subcatchment 9), detailed facilities location are shown in Figure 4.

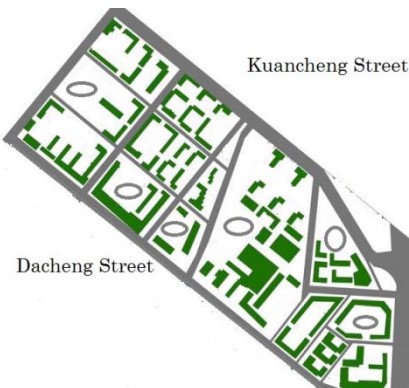

**Figure 4.** Planning for Sponge City (map scale is 1:15,000, green areas are green roofs and grey circles are sand ponds).

*3.2. Initial Costs for Drain Replacement Project and Sponge City Project*

3.2.1. Initial Costs for Drain Replacement Project

- Construction fee Based on 'Quota' and the existing studies (Wang et al., 2008), unit cost was calculated as 4846 RMB/m (inflation rate was considered as 2.5%/year in 10 years). Project lengths of Kuancheng and Dacheng street are 895 m and 912 m respectively, total project period is 70 days.
- Maintenance fee According to Harbin City urban road maintenance standard (Table 1), the unit maintenance fee was obtained as 41.36 RMB/m.

**Table 1.** Unit maintenance fee by drain diameter

| Specification (mm) | Amount | Unit |
| --- | --- | --- |
| $DN\Phi < 600$ | 20.54 | RMB/m·year |
| $600 \leq DN\Phi < 1000$ | 21.83 | RMB/m·year |
| $1000 \leq DN\Phi < 1500$ | 41.36 | RMB/m·year |
| $DN\Phi \geq 1500$ | 38.91 | RMB/m·year |

### 3.2.2. Initial Costs for Sponge City Project

- Construction fee According to 'Indices', unit container green roofs and sand pond costs are obtained as 39,120 RMB/100 m$^2$ and 28,878 RMB/10 m$^3$, the project period is 90 days.
- Maintenance fee Unit maintenance fee was obtained from completed sponge city sample project (Qunli Healthy Ecological Garden) in Harbin [19]. The maintenance fees for this project were calculated as 702,000 RMB in total.
- Other categories In the Sponge City project, positive environmental impacts are anticipated. According to the 'manual', 20% of annual precipitation could be stored (annual average precipitation of Harbin is 569 mm). Therefore, 6885 m$^3$ storm water could be harvested, which can reduce carbon emission from the equivalent tap water production; based on existing study (Bianchini et al., 2012), green roofs can save 4.3 kWh /m$^2$ in cooling energy consumption. Based on top water and electricity emission factors in China being 0.91 and 0.785 kg-CO$_2$/tonne respectively (Lin et al., 2018), reductions of 6.27 and 197 tonne-CO$_2$ emission annually can be calculated, respectively.

### 3.3. External Costs for Drain Replacement Project and Sponge City Project

### 3.3.1. External Costs for Drain Replacement Project

- Carbon emissions from material production Main material in drain replacement project is concrete. According to industries standard in China, concrete carbon emission factor in life cycle is calculated as 1.33 tonne-CO$_2$/m (Lin et al., 2018), which had considered the production, transportation, and disposal processes.
- Carbon emissions from equipment Based on 'Quota' and electricity emission factor in China, unit data was calculated as 0.107 tonne-CO$_2$/ m.
- Increasing emissions of CO and NOx by traffic regulation In transport assignment models the common way to describe the relationship between travel time and traffic flows is the BPR function [20], as

$$T_r = TF_r \left\{ 1 + \alpha \left( \frac{f_\alpha}{Q_\alpha} \right)^\beta \right\} \tag{3}$$

$T_r$: the total travel time on link $r$

$TF_r$: the free travel time on link $r$

$f_\alpha$: refers to the traffic volume and is introduced as 2200/h for one lane

$Q_\alpha$: the travel capacity of link $r$, $\alpha$, and $\beta$ are the traffic/delay parameters. It was assumed that left and right of Equation (3) were divided by travel length, and another equation was obtained.

$$V_r = \frac{VF_r}{\left\{ 1 + \alpha \left( \frac{f_\alpha}{Q_\alpha} \right)^\beta \right\}} \tag{4}$$

where $V_r$ is the travel speed in link $r$, $VF_r$ is the free travel speed in link $r$, 60 km/h was introduced in this paper. $f_\alpha$, $Q_\alpha$, $\alpha$, and $\beta$ were the same with above.

In this paper, we tried to use BPR function to estimate the CO and $NO_2$ emission changes accomplished with traffic flow changes.

The detailed calculation was shown by the following:

a.　The link from Wenhua to Nanli was set as link A (Kuancheng), and its length is 0.895 km; the link from Zhang kou to Daoli was set as link B (Dacheng), and its length is 0.912 km. Link A and B was simulated as a target link.

b.　Calculation of $\alpha$, $\beta$. In order to calculate conveniently, here set $\beta$ as 1, according to the measured traffic volume and travel speed, $\alpha$ was calculated (see Table 2).

c.　In a three-lane street, one lane was blocked, travel capacity from morning 7:00 to 19:00 was decreased from 79,200/day to 52,800/day. We assumed that the traffic volume would not change with or without street blocks. Therefore, the travel speeds with a blocked lane were calculated by BPR function.

d.　Based on vehicle emission coefficient from Japan National Institute for Land and Infrastructure Management (NILIM) (we did not find Chinese emission coefficient standard) [21] (p5–43 and p5–50), extraordinary emission of CO, and $NO_2$ per vehicle per day due to the decrease of vehicle average speed was calculated.

e.　Multiplied by traffic amount in target time zone (from morning 7:00 to 19:00) and project duration (70 days), the increasing emission of CO and $NO_x$ are calculated.

**Table 2.** Traffic data in two links (target traffic time zone is from 7:00 a.m. to 7:00 p.m., 12 h, in one direction)

| Subjects | Link Kuancheng ($\alpha$ = 4.48, $\beta$ = 1) | Link Dacheng ($\alpha$ = 3.92, $\beta$ = 1) |
|---|---|---|
| Travel delay distance | 0.895 km | 0.912 km |
| Amount of traffic | 51,199 | 39,512 |
| Measured average speed | 15.4 km/h | 20.3 km/h |
| Predicted average speedby BPR function mentioned above [20] | 11.2 km/h | 15.3 km/h |

There are six lanes in both Kuancheng and Dacheng Street, and one lane in one direction are assumed to be blocked because of drain replacement. Data measured by experimented and calculated are shown in Table 2, and the increasing emission of CO and $NO_x$ are calculated as 125 tonnes and 0.179 tonnes.

- Noise and vibration pollution costs A basic construction area noise and vibration levels are 85 dB and 75 dB, the comfortable noise maximum value and natural vibration level are 55 dB and 65 dB. Difference can be calculated. Numbers of people affected can be calculated by population density (315/ha) and affected area (20 m away from the streets, 40 m both sides). Unit monetary costs of noise and vibration pollution are 0.75 and 0.56 (RMB/dB, people, day) respectively, excavation time is 10 days.

- Travel delay (time loss) In traffic simulation section, delay time for one vehicle could be calculated. Multiplied with traffic amount and time value 20 RBM/h (adjustment index by 2.5% based on Chinese study in Table 3), time loss could be calculated as 26,631,990 RMB.

**Table 3.** Unit time value

| Matthews | Iwashita | Wang |
|---|---|---|
| 2015, US | 2011, Japan | 2008, China |
| 80 RBM/h | 150 RMB/h | 15.6 RBM/h |

3.3.2. External Costs for Sponge City Project

- Carbon emission from materials production Material utilized in the project mainly includes sand, concrete, and steel, and the volumes are calculated as 4709 $m^3$, 456 $m^3$, and 38.991 tonnes respectively by 'Indices'. Their life cycle emission factors were obtained from industries standard as 5.97 kg-$CO_2$/$m^3$, 1.33 tonne-$CO_2$/ $m^3$, and 2.62 tonne-$CO_2$/ tonne respectively.
- Carbon emission from equipment Based on 'Indices' and electricity emission factor, unit data was calculated as 0.138 tonne-$CO_2$/$m^2$.
- Other categories In the Sponge City project, external positive environmental impacts are also anticipated. Unit carbon reduction data for *Sedum lineare* average carbon absorbing capacity was derived from previous researches, emission volume was estimated as 17.7 tonne-$CO_2$/ha; air quality improvement is related to the mitigation of nitrogen oxide (NOx). Unit data was derived from previous studies as 0.326 tonne-NOx/ ha.

*3.4. Results*

In order to reduce discharge of 3049 $m^3$ storm water over flow, drain replacement and Sponge City were designed and constructed, initial unit and external unit costs were calculated and shown in Figure 5 respectively (COx and NOx monetary cost are not included).

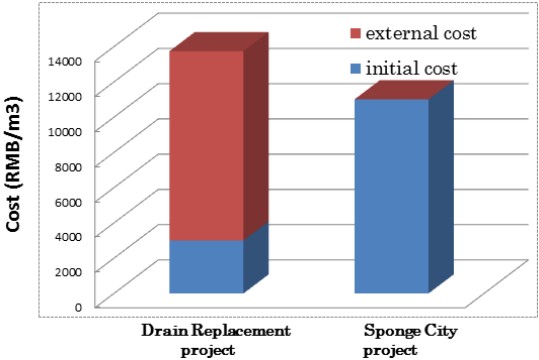

**Figure 5.** Comparison between two projects.

The figure above shows that the cost is different with or without considering external cost. Without considering external costs, the unit cost of the drain replacement project is smaller than the unit cost of the Sponge City project; conversely, with considering external cost, the former is bigger than the latter, which has shown that Sponge City is sustainable to employ in urban environments.

**4. Discussion**

Overflow calculation in methodology presented in this paper is based on runoff coefficient, so accurate land cover and land use condition data are necessary and helpful to improve calculation accuracy.

Owning to precipitation data limitation, we only set two years' frequency (return period), this could lead to rainfall underestimating which could causes designed discharge (drainage system) and retention capacity (Sponge City) insufficient. Besides, as mentioned above, methodology presented in this paper could be conducted when precipitation is not extreme in volume. In fact, many researchers have concluded that Sponge City (green roofs) has its limitations in mitigating urban flooding. For instance, Elizabeth et al. showed that by building green roofs with the layer thickness between 50 and 150 mm, the runoff could be prevented if the precipitation is less than 25 mm and if precipitation is over 25 mm/h, the overflow prevention is not obvious in Auckland, New Zealand [22]; Mentens et al. also mentioned that it is clear that roof greening alone will never fully solve the urban runoff problem and it needs to be combined with other runoff reduction measures [23]. Conventional open trench

activities result in disruption to urban areas through road closures, traffic delays, time loss of access to homes or business, as well as noise and vibration disruption for everyone in the surrounding area. Trenchless technologies are a replaceable solution to China's buried infrastructure. However, numerous challenges are present when promoting trenchless technologies in China, including minimal local engineering knowledge, lack of trained contractors, lack of specifications, and system impact concerns raised by government owners. This paper presents a countermeasures comparison in which buried infrastructure replacement has been constructed in an open cut way.

## 5. Conclusions

The urban drainage profession has undergone significant change over the last several decades, moving from an approach largely focused on flood mitigation and health protection to one in which a wide range of environmental, sanitary, social, and economic considerations are taken into account. The profession has thus developed and adopted new terms to describe these new approaches and is likely to continue to do so, as the transition to a more sustainable and integrated approach occurs.

Using combined sewer systems to handle excess stormwater runoff is common in older urban areas. As urban population density increases and more demand is placed on infrastructure, combined sewer overflow events happen more often and cause serious environmental problems. Recently, green infrastructure (Sponge City) has been integrated with existing gray infrastructure (drainage pipes) to reduce combined sewer overflow events. However, the quantization analysis on the substitution between these two infrastructural approaches is insufficient.

From calculation above, we have come to the conclusion that only considering initial cost, compared with drain replacement project, although Sponge City can reduce 198 tonnes of $CO_2$ emission annually, its unit cost is much more than the drain replacement project unit cost; considering external costs, Sponge City is not only competitive from an economical perspective, but from an environmental perspective in the third year its carbon neutrality is expected to be reached, which means from the fourth year Sponge City could function as a carbon emission reduction factor and the Sponge City project could also absorb 1.9 tonnes of NOx annually, which shows Sponge City is competitive.

However, precipitation data presented in this paper is not long enough in years and could mean the designed discharge would be insufficient, which could cause infrastructure to lose its overflow controlling function. Going forward, after collecting a larger range of precipitation data, we will calculate situations wherein the retention capacity of Sponge City is exceeded, in which both systems may be considered in their complementarity: an upgraded drainage system plus green infrastructure. Our present work provides a framework to estimate the possibility of partial substitution between a traditional sewer drainage project and Sponge City in a non-extreme rainfall situation.

This paper presents a framework to compare two infrastructural approaches to urban waterlogging at a community scale and concluded that although Sponge City is a new contermeasure to mitigate urban waterlogging, its total social cost throughout its life cycle is lower than a conventional drain replacement project.

**Author Contributions:** Supervision, T.M.; Writing—original draft, X.F. All authors have read and agreed to the published version of the manuscript.

**Funding:** This research received no external funding.

**Acknowledgments:** This study has been supported by Toru Matsumoto, The University of Kitakyushu.

**Conflicts of Interest:** The authors declare no conflict of interest.

## Abbreviations

| | |
|---|---|
| IUWM | Integrated Urban Water Management |
| LID | Low Impact Development |
| SUDS | Sustainable Urban Drainage Systems |
| WSUD | Water Sensitive Urban Design |

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
