# Peer review of "Comparative Analysis on Urban Flood Countermeasures Based on Life Cycle Thinking: A Comparison between Enhancing of Drainage Capacity Project and Sponge City"

_environments, doi:10.3390/environments7070051_

Round 1

Reviewer 1 Report

Overall this work has merit and will be of interest to readers. 

The English is understandable, but does not flow well - it needs to be proof read. 

The paragraph at the top of page two should also make reference to SUDS (Sustainable Urban Drainage Systems). Tim Fletcher published a paper which reviewed SUDS\LID\WSUD etc - referencing this here would help.  

Figure 3 is not clear and adds little. A better overview of the case study area is needed. 

WSUD (etc) are presented as a range pf measures, but the deployment strategy / design approach is also important.  Can more detail be provided on this? 

Figure 4 is helpful, but what is the scale? 

ton is used instead on tonne

The analysis produces interesting results, but greater emphasis needs to be placed on uncertainty. 

The conclusion needs to consider to what extend these results are transferable. 

Reviewer 2 Report

The topic is interesting. Nevertheless, the manuscript needs some further improved before to be accepted for publication. In general, there are still some occasional grammar errors through the manuscript especially the article ‘’the’’, ‘’a’’ and ‘’an’’ is missing in many places, please make a spellchecking in addition to these minor issues. The reviewer has listed some specific comments that might be helpful of the authors to enhance the quality of the manuscript further.

  • The abstract should be improved further, describe shortly objectives, methods and main findings.
  • List of abbreviations is needed.
  • The objectives are not explicitly stated in the introduction.
  • The authors need to enrich further the background; please highlight some devastating historical cases in China would suggest considering the following literature << Human–Environment Natural Disasters Interconnection in China: A Review >> you may review other additional relevant references as well.
  • Methodology limitations should be mentioned.
  • Data acquisition and resolution also.
  • Figure3, should be in section 2, and please improve the quality, north arrow, scale bar, present it professionally.
  • Description of land cover and land use is needed, a map also.
  • Description after section 3 up to section 3.1 should be moved to methods, that does not result.
  • The results section should be elaborated more; there very few results.
  • The discussion should provide a summary of the main finding(s) of the manuscript in the context of the broader scientific literature, as well as addressing any limitations of the study or results that conflict with other published work.
  • The conclusion should be elaborated more; please put some quantitative findings.
  • Please check the references in the text and the list; some of them are not according to the journal style.
  • All references should be updated, please consider some relevant additional references.

Reviewer 3 Report

Dear authors,

Please, read the attached file. Some improvements and clarifications are necessary. 

Best regards,
